# “I Assumed It Would Be Somebody Who Had a Stroke That Was Doing This”: Views of Stroke Survivors, Caregivers, and Health Professionals on Tailoring a Relaxation and Mindfulness Intervention

**DOI:** 10.3390/healthcare11030399

**Published:** 2023-01-31

**Authors:** Thomas Atkinson, Emma Brown, Georgina Jones, Karen Sage, Xu Wang

**Affiliations:** 1Psychology, Leeds School of Social Sciences, Leeds Beckett University, Leeds LS1 3HE, UK; 2Applied Clinical Research, Manchester Metropolitan University, Manchester M15 6BH, UK

**Keywords:** patient and public involvement, PPI, stroke, secondary prevention, relaxation, mindfulness, tailored techniques, prompts and cues, behaviour change

## Abstract

Stroke survivors and informal caregivers experience high levels of stress and anxiety, linked to heightened risk of secondary stroke in survivors. Relaxation and mindfulness could reduce stress and anxiety; being most effective when tailored to the target populations. Aims of the PPI include to: (1) consult on possible alterations to an existing relaxation and mindfulness intervention, delivered via YouTube/DVD and (2) discuss relevance and preference of prompts and cues designed to facilitate the daily practice of the intervention. Eleven UK PPI contributors were consulted during 2020: four stroke survivors (F = 2, M = 2), three caregivers (F = 1, M = 2), and four HCPs (F = 4) (range = 23–63 years). Contributors watched the existing intervention and provided feedback via online discussions. Transcripts were analysed using thematic analysis. Five themes were identified, highlighting several necessary alterations to the intervention: “Who represents the stroke population?”; “The paradox of age”; “Specifically selected language”; “Visual presentation of the intervention”; and the “Audio qualities”. Contributors ranked the prompts and cues in order of preference with setting alarms and email alerts as the most popular. The PPI consultations resulted in several alterations enabling a revised version of the intervention. Including a PPI consultation at an early stage of the research improves the relevance and appropriateness of the research. The revised intervention is more representative of the stroke population thus more likely to be practised by survivors and caregivers, which will enhance the extent of effectiveness, reducing the risk of a secondary stroke.

## 1. Introduction

### 1.1. Background

The prevalence rate of post-stroke anxiety has been shown to be between 18.7% and 24.3%. Moreover, these rates were not shown to reduce in up to 24 months after a stroke [1]. Post-stroke anxiety is related to a poor quality of life and depression [2] as well as being associated with a lack of confidence in social participation, loss of identity, and negative impacts on daily living [3,4,5], highlighting the severity of the issue. Moreover, research has demonstrated that psychological stress can have a negative impact on the recovery of stroke survivors and can lead to poorer long-term outcomes [6,7]. Psychosocial stress has been identified as a secondary stroke risk factor [8]. Consulting a stroke-related PPI group to tailor an intervention to reduce stress and anxiety will therefore encourage engagement from the end users and have an enhanced possibility for the intervention to work, thus reducing the risk of secondary strokes [9].

Psychological interventions designed to reduce anxiety and distress in stroke survivors are limited [1]. However, promising results have been demonstrated regarding the use of mindfulness and relaxation techniques. Mindfulness is defined by Kabat-Zinn (2003) as a particular way of paying attention that is intentional and focuses on the present moment. This process occurs non-judgmentally which means to focus on a particular present experience while distancing thoughts and emotions that may occur [10]. In terms of reducing stroke-related anxiety and stress, Jain et al. (2007) suggests that mindfulness can help by decreasing rumination and increasing attentional control [11]. 

Relaxation techniques encompass multiple techniques such as progressive muscle relaxation and autogenic relaxation. Relaxation is generalised as a cognitive/behavioural practice which focuses on eliciting the relaxation response aimed at counteracting the body’s stress response through decreased arousal [12] generated through repetitive physical or mental activity while not engaging with distracting thoughts [13]. Relaxation and mindfulness share overlapping elements but also differ in their mechanisms. However, they produce similar outcomes such as reducing stress and anxiety, thus it would be useful to have both in one intervention which aims to reduce stress and anxiety for stroke survivors and caregivers. It is also recommended that an intervention can include multiple techniques and components as opposed to one with a single component [14]. 

Golding et al. (2015) administered a self-help autogenic relaxation intervention to stroke survivors in a pilot randomised controlled trial. Participants practised relaxation techniques delivered via a CD for one month and a significant reduction in anxiety was observed when compared with the control group. Moreover, after three months, 40% of participants no longer met the clinical level of anxiety [15]. After a one-year follow-up, results indicated that these reductions in anxiety had been maintained [16]. Lawrence et al. (2013) conducted a systematic review into mindfulness-based interventions following a stroke. Results indicated benefits in depression, anxiety, perceived health, blood pressure, and mental fatigue [17]. Whilst authors reported no evidence of harm in using mindfulness interventions, we should acknowledge that a systematic focus is rarely applied to individual level data, which is required to explore this further [18]. 

However, it is not only the stroke survivor that is impacted by stroke, but it is also well documented that caring for a survivor can have detrimental effects on physical and mental health [19,20]. An informal caregiver is defined as individuals who are unpaid and provide care for an individual who due to disability and/or illness struggle to complete daily activities [21]. One study found that caring for a stroke survivor results in increased caregiver strain [22] and a pooled prevalence rate of 21% for anxiety symptoms [20]. Demers et al. (2021) explored the feasibility of an online mindfulness intervention in a small sample of stroke survivors and stroke caregivers. Participants did report subjective benefits; however, no change was found for self-reported psychological wellbeing measures [23]. Therefore, further research into the benefits of mindfulness for the target populations are required. Another study by Yilmaz et al. (2019) looked at progressive muscle relaxation on informal caregivers of stroke survivors in a randomised controlled trial [24]. The results demonstrated a reduction in depression scores as well as caregiver burden; however, when compared with the control group, they did not differ significantly. A knowledge gap exists in using relaxation and mindfulness interventions for informal stroke caregivers, and larger more in-depth studies are required to explore how this can be improved. 

Patient and public involvement (PPI) in health research is beneficial to research and to the individuals contributing [25]. PPI can be defined as an activity that has been undertaken by or in partnership with members of the public in contrast to research conducted for or about them [26]. PPI has been shown to make a difference to the overall research outcomes when incorporated into various elements of the research cycle, ranging from prioritisation of topics, development of research questions, dissemination, and follow up processes [27]. Furthermore, Price et al. (2017) suggested that the value of the health research to the end user, quality, content, and consistency can be improved with the inclusion of patients and the public [28]. 

Liberati (2011) highlights that there is often a mismatch between what a patient wants or needs compared with what clinicians/researchers focus on [29]. Highlighting that both researchers and PPI contributors do not have the relevant expertise or scope to implement effective change in research individually. However, when combined and both areas of expertise are represented in the research cycle, effective outcomes can be achieved. Indeed Chalmers et al. (2014) identified that one factor in wasted research is when the needs of the end user are ignored [30]. People are more likely to adhere to and practise techniques that are tailored to their needs and preferences [31]. Harrison and Palmer (2015) suggest that the evidence base for the involvement of stroke survivors in PPI is limited [32]. However, they found that when survivors and caregivers were included the process was beneficial to them in several ways, including the concept of giving something back or doing something useful. They also found that there was a perceived benefit to the research process in the form of providing a different perspective. Gibson et al. (2012) extend upon this and highlight that the knowledge of lay persons and professionals are both vital, and it is important in PPI that they are treated equally [33]. 

Interventions in digital format have been shown to benefit conditions and health behaviours such as depression, asthma, and smoking cessation [34,35,36]. However, interventions regularly report small effect sizes, possibly caused by infrequent use [37]. Moreover, this conclusion was supported by Wang et al. (2019) when conducting a feasibility study, which identified that stroke survivors found it difficult to incorporate mindfulness and relaxation techniques into their daily routines, which resulted in the intervention not being completed to the desired frequency (i.e., daily) [38]. The importance of daily practice is highlighted by Parsons et al. (2017) who found a small to moderate association between home practice and treatment outcome [39].

Prompts and cues are one way in which engagement with a digital intervention could increase frequency of practice. Michie et al. (2013) defines prompts and cues as the introduction of an environmental or social stimulus with the intention of prompting or cueing a desired behaviour, which in this case would be daily practising of the intervention [40]. Indeed, frequent practicing of mindfulness and/or relaxation can be viewed as a behavioural and lifestyle choice [41]. The prompts and cues are normally targeted to occur at the time or place of practice. These can include using stickers, fridge magnets, or daily reminders through phone alarms or emails. A systematic review exploring technology-based prompts on digital interventions’ effectiveness found a small to moderate effect on increasing engagement [42]. However, the included studies had small sample sizes and did not explore the characteristics of each behaviour change technique or the effect of individual difference and preference. Therefore, more research is required into their use and specifically the tailoring of them to the target population. This was also explored in the current PPI. 

Bellg et al. (2004) suggest that for any proposed intervention aimed at health behaviours to be successful, it must go through three stages [43]. First, the intervention must be successfully delivered to the participant in a way that makes the techniques clear. The individual who is completing the intervention must then fully understand what is expected of them and know exactly what they have to do. Finally, they have to complete the techniques how they are meant to be carried out. From this, research has demonstrated that successful and sustained behaviour change from an intervention depends on the individual’s ability to self-enact and employ behavioural change techniques individually [44,45]. This is particularly important in the stroke populations as the impact of stress on the cognitive and psychological domains of stroke survivors can lead to poor motivation which influences the individual’s engagement with rehabilitation and up taking interventions [46]. 

### 1.2. Description of the Current Intervention

The current intervention consists of four relaxation and mindfulness techniques filmed as a video and delivered in a digital format via YouTube link or DVD, if required. Users are asked to practise the intervention frequently, on a daily basis, for a set period of time of 10 min, to gain maximum benefit. The original video intervention had been adapted previously to be more accessible to stroke survivors, particularly those with aphasia. Details of how the intervention has been specifically tailored previously can be found in Wang et al. (2019) [38]. The focus of the current PPI was to discuss whether the original adaptations had been successful and if each of the techniques were accessible for the stroke populations. 

### 1.3. Aims of the PPI

The aim of the PPI was to (1) consult contributors to determine potential alterations to an existing relaxation and mindfulness intervention; and (2) capture perspective and preference on a range of prompts and cues to facilitate daily practice of these techniques.

## 2. Materials and Methods

### 2.1. PPI Framework

In accordance with the PPI framework conceptualised by Tritter (2009) [47], the current PPI model selected is proactive, collective, and direct. Our PPI contributors: stroke survivors, informal stroke caregivers, and health care professionals (HCPs) were actively shaping a future mindfulness and relaxation intervention that is tailored to the stroke populations, improving its acceptability and accessibility. This manuscript has been written in alignment with the GRIPP 2 framework [48] (Appendix A).

### 2.2. PPI Contributors 

Due to COVID-19 restrictions several online recruitment methods were used in 2020. Stroke survivors, informal caregivers (hereafter will be referred as ‘caregivers’ throughout the paper), and HCPs who work closely with the stroke community were recruited through local stroke support groups in North and West Yorkshire and through the researcher’s (TA) university Service user and Carer group. Gatekeepers for stroke support groups were contacted on social media and asked to advertise the PPI. Furthermore, advertisements were placed on TA’s social media along with utilising networking opportunities. 

A total of 11 contributors participated in the PPI. They consisted of four stroke survivors (F = 2, M = 2), three stroke caregivers (F = 1, M = 2), and four HCPs (F = 4). Stroke survivors were required to be: (1) over the age of 18; (2) a UK resident; (3) had a confirmed stroke via medical diagnosis; and (4) living in the community (i.e., no longer hospitalised). Two survivors also had a confirmed diagnosis of aphasia. Stroke survivors were aged between 23–63 years of age. A brief description of the contributors’ social demographic background can be found in Table 1. 

Caregivers were aged 18 years old and over and had experience of caring for an adult stroke survivor. Caregivers can include spouses, parents, or friends, but will exclude professional paid caregivers. Caregivers were aged between 26–63 years of age. The length of time that they had been a stroke caregiver ranged from 3 to 15 years. 

Health care professionals were required to be over the age of 18 and to have worked closely with stroke survivors in a professional capacity. HCPs were aged between 23–53 years of age with between 1–19 years of experience within their current role.

### 2.3. The Current Intervention

The overall intervention video lasted 14.24 min consisting of two mindfulness techniques and two relaxation techniques. Mindfulness and relaxation are distinct interventions. However, there is some overlap in practice, as which technique(s) might be better considered as relaxation or mindfulness. For example, breath watch has been considered a relaxation technique by Benson and Klipper (2000) [49]. Body relaxation is based on the principles of autogenic relaxation which had been tailored to stroke survivors’ needs in previous studies [38,50]. The four techniques were selected and tailored for use with stroke survivors and those with communication difficulties [38]. A brief description of each technique can be found in Table 2. 

The video starts with a brief introduction that covers what relaxation and mindfulness techniques are and why they are beneficial. Each of the techniques are demonstrated by a professor who talks through each step of the different techniques using various graphics and key words displayed on the screen to aid their understanding. After each of the techniques are demonstrated, the user is instructed to practise each technique in turn and the screen becomes blank for 1.5 min for the technique to be practised. After the allotted time, a gong sounds to indicate that the practice time is over. The demonstrator reappears on the screen and moves to the next technique. 

### 2.4. Measures and Apparati

To ensure accessibility, all documents used in the PPI were aphasia-friendly. This involved creating specific aphasia-friendly forms that included visual information to guide understanding. Moreover, information sheets were also recorded by TA so that contributors could listen to rather than read the information if they prefer. Individuals with aphasia are often excluded from research studies [43] and by presenting documents in a multi-sensory format the authors aimed to be as inclusive as possible. The research team were a multidisciplinary team including a professor specialised in speech and language science. All contributor facing documents were revised by the research team and placed in multiple formats to maximise accessibility. 

A demographic questionnaire was designed that covered seven questions for all PPI contributors to complete. These included: age, gender, ethnicity, marital status, educational background, and which contributor group they belonged to. Stroke survivors were asked how many strokes they had experienced and when the most recent one had occurred. Stroke caregivers were asked how long they had been a caregiver for, if they had worked before becoming a caregiver and if so, to provide their profession. Moreover, if they were currently employed. HCPs were asked for their profession and how long they had been in that role. 

An interview topic guide was used to explore PPI contributors’ views and feedback on the intervention and the proposed prompts and cues. The topic guide covered questions about the overall intervention such as: “How long after stroke would be the best time to deliver the intervention?”. The specifics of the video intervention such as: “Are the relaxation and mindfulness techniques clearly explained?”. Furthermore, questions related to the prompts and cues: “What prompts do you think would be most relevant/useful to you?”. A speech and language therapist (KS) was consulted to ensure that the questions were appropriate and that all language and sentence structure was aphasia-friendly. 

Contributors were provided with a list of the 13 possible prompts and cues. These included: Fridge magnets, stickers, text message reminders etc. (see Table 3). Contributors were informed that they could use any individual or combination of the prompts and cues in practise, however, for the purposes of this PPI, contributors were asked to rank them in order of what they perceived to be most beneficial. There was also a section to provide any other suggestions that they felt could be beneficial in terms of prompting them to practise.

### 2.5. Procedure

Ethics approval was granted by the authors’ institutional research ethics committee. When gaining consent, it was vital that contributors were fully informed. All contributors were provided with a consent form that they could fill out electronically or via a hardcopy. Stroke survivor contributors were given the option to provide consent verbally if they preferred, which was then recorded. 

Qualitative interviews were chosen as the format for the PPI consultation. Qualitative methods contribute to gaining a more detailed knowledge of the perception, understanding and experiences of individuals or groups amongst the population [51,52]. Due to COVID-19 restrictions being implemented at the time of the PPI, it was deemed that individual interviews conducted online would be most appropriate. For stroke survivor contributors it was designed so that they could complete the PPI in two 45-min sessions with regular breaks, when required, to avoid fatigue. However, this was flexible, and survivors could choose how long they needed and could schedule meetings for the next day, or whenever they felt ready. Stroke caregivers and HCPs completed the interview in one session lasting approximately one hour. Contributors were offered an online conversation or telephone call depending on what they were most comfortable with. Online platforms consisted of using either Google Meet or Microsoft Teams. All meetings were recorded and transcribed verbatim. 

Prior to interview commencement, all PPI contributors were offered the intervention via a YouTube link or DVD depending on technological ability or preference. They were encouraged to watch the demonstration video and attempt to practice each of the techniques as if completing the intervention themselves. Contributors were also sent a list of proposed prompts and cues intended for use in future research to facilitate the desired frequency of practice. Contributors were provided with this to familiarize themselves and to avoid overwhelming them during the discussion, which already had a large amount of information to cover. 

When discussing the prompts and cues, the researcher displayed their screen and read them out to contributors, if required. This offered the opportunity to answer any queries before moving onto discussing the questions on the topic guide and helped the contributors to familiarize themselves once more. Finally, contributors were asked to rank their top five prompts and cues that they thought would be most beneficial to them in facilitating regular practice of the relaxation and mindfulness intervention. TA used this opportunity to gain more information on why they preferred these five. 

### 2.6. Analysis

PPI members are part of an advisory group for the doctorate project. All the PPI contributor sessions were recorded and then transcribed verbatim with names removed to maintain anonymity for the purpose of this publication. Thematic analysis was then conducted based upon the detailed procedure outlined by Braun and Clarke (2021) [53] using the following steps: familiarizing yourself with the data set, coding, generating initial themes, developing, and reviewing themes, refining, defining, and naming themes; writing up. The 15-point checklist was used to inform good practice [53]. 

## 3. Results

### 3.1. The Intervention

The results of the thematic analysis concerning PPI contributors’ views on the current intervention was split into five themes: “Who represents the stroke population?”; “The paradox of age”; “Specifically selected language for the stroke population”; “Visual presentation of the intervention”; and the “Audio qualities” of the intervention. Each theme is discussed in further detail below.

#### 3.1.1. Theme One: Who Represents the Stroke Population?

The first thing to note is that the current intervention is designed to be a self-help method. Therefore, it is vital that stroke survivors take ownership of this and fully engage with it. The intention is that stroke survivors and caregivers can practise the techniques independently and do not need to rely on support from HCPs or other people to complete them. This was commented on very positively by a speech and language therapist in the PPI:

“We would always promote independence, when we do therapy it’s always try and do it independently because in speech therapy, they are communicating for themselves and they are taking control of it, whereas that (practising the intervention) is taking control of their own wellbeing isn’t it and I think that helps. That energises them because they are getting back some independence”(HCP4)

Although, this is a very positive appraisal of the intervention, it is not without criticism and this builds into the first theme that looks at who represents the stroke population in the intervention video. From this it became clear that like the definition of PPI, a research activity should be undertaken by or in partnership with members of the target population, therefore, interventions should follow the same school of thought.

“I assumed it would be somebody who had had a stroke that was doing this.”(HCP4)

Despite the fact that the intervention had undergone efforts to make the selection and adaptation of techniques suitable for stroke survivors, the demonstration of techniques in the video was still presented by an able-bodied person (i.e., a professor in health psychology). This issue of representation was developed further by the quote below.

“Able-bodied presenters, it’s like seeing the adverts on TV for Stannah stairlifts and they’re all able-bodied actors.”(SC1)

The phrase “Stannah stairlifts” refers to a company who advertise their stairlifts on TV. In the advert an able-bodied demonstrator shows the viewer a stairlift in action. It makes the product look incredibly easy-to-use but it is very apparent that it is not the end user demonstrating the product. The above quote links to the current intervention video in explaining that it is an able-bodied presenter who is demonstrating how to use techniques, for stroke survivors.

“I think it’s important to make it clear to the receiver, that the qualifications and abilities of the presenter, erm it’s not, it’s like someone giving a weather report and not being a meteorologist.”(SC1)

Again, this shows that individuals want the demonstrator to appear to know what they are talking about or visually have experience of the topic.

The demonstrator in the video was immediately identified as an academic. During the PPI discussion, one of the topic guide questions focused on if the demonstrator should be an individual who could be clearly identified as belonging to the medical profession (i.e., a credible source). The initial thinking of asking this question was that this individual would display the perceived skill set and authority that would be required to enhance engagement with the intervention.

This concept of perceived authority is demonstrated by SC3 when discussing a stroke survivor’s engagement with a physiotherapist. He commented that “if I told her to do something she wouldn’t… but when the physio told her… she was all over it and wanted it…”. This talks about a caregiver trying to get a survivor to complete some form of exercise and not having success. Once a person who the survivor saw as having some authority regarding the issue suggested it, they became engaged in the activity. However, a medical professional may not be the best option as stroke survivors have often met a vast amount of health care professionals and could become burnt out. For example, a neuro physiotherapist (HCP4) commented that “some stroke survivors might be so fed up with being told by health care professionals what they need to do, or they might almost rebel against that in a way”.

Therefore, if it is possible to keep this perceived authority around the topic but move away from health care professionals and a clinical setting by replacing it with lived experience, this could be a beneficial way of increasing uptake and engagement as this lived experience is irreplaceable. For example, caregiver SC1 fed back that “it doesn’t matter how good the students are at acting, it’s not the same as talking to SS2”. Here, the caregiver is talking about health care students practising clinical skills on actors as part of their courses. He states that even if they are very good actors portraying a stroke survivor, it will not be the same experience as actually working with a real stroke survivor. This again highlights that by having a stroke survivor as the demonstrator it adds to that sense that you know what stroke survivors and/or their caregivers are going through. Additionally, you can tell when someone is telling you something from what they have learnt compared with when they have lived through an experience. This adds to the ownership of an intervention as something that is for stroke survivors rather than something that has been repurposed for stroke survivors.

For stroke survivors, they wanted the demonstrator in the video to have a clear understanding of what they have experienced and the difficulties that they face. This is especially true when that individual is prescribing or advising them to do something. This could be seen as authority of experience where a survivor could be more inclined to listen if they feel the individual understands what it is like to experience a stroke compared with someone who has not.

#### 3.1.2. Theme Two: The Paradox of Age

Because there was quite an extreme range of ages within the stroke survivors who contributed to the PPI, a contradiction between the age of the demonstrator appeared. 

Stroke can occur at any age but with higher volumes of the elderly being the traditional target population, interventions and resources are normally aimed at them. HCP4 commented that younger stroke survivors might “struggle having somebody older telling them or going through this video with them or listen to their voice… but I just think for something like this you might need to have somebody’s voice sounding a little younger than this particular man’s voice.”

This highlights a gap in the resources available and a very difficult issue to address when developing interventions. With such an age difference within the population, picking the right demonstrator or facilitator for an intervention becomes even more vital. This point of age goes further as the survivor goes on: 

“… if you had someone who was 18, yes they might have had a stroke, but people might question, older generation typically might question I don’t believe they have had a stroke and that’s how I generally find things.”(SS4)

Even though a younger survivor has been through similar experiences to an older survivor, the perception of their age is something to be considered. Here, the younger survivor reflects on their experience when talking about people questioning if they have had a stroke or not. This judgement of age and lack of life experience is presented by a health care professional below. 

“Yes, I think, yes if, unfortunately if you have someone looking really young and very fit, sometimes it can have a negative effect because you are just judging from the video… I do think people sometimes find it hard if they haven’t got some life experience behind them, so if they have just come out of Uni but they are telling me to relax, they don’t know what I have been through.”(HCP3)

Again, this highlighted that even if you do not know what an individual has experienced or the skill set that they possess, a judgement may be made about their age and what experiences they may have. Moving forward, it is vital that these issues are addressed. This could be either targeting differences explicitly and having multiple options of demonstrators to attempt to engage different populations or by addressing these judgements. This could be possible by having a short introduction which explicitly states the experiences of the demonstrator adding to the representation, where visual cues of experiences may not be clear.

#### 3.1.3. Theme Three: Specifically Selected Language for the Stroke Population 

The benefit of working closely with PPI members can be seen in the use of language in the intervention, especially when involving techniques that require multiple elements. 

“… I just think it could be aphasia-friendly, but at the moment it is not because of the really complex sentences he’s using and the really confusing subtitles…”(HCP2)

“… I think there is too much information, abstract language; I think you just need short simple sentences…”(HCP1)

In order for interventions to be aphasia-friendly, the language used in them needs to be appropriate. In the current intervention there were words such as “equilibrium” that were not essential to the meaning of the sentence.

Another issue specifically regarding the language and its suitability for stroke survivors was raised below regarding the systematic relaxation of different body parts. Here, hemiplegic limbs must be considered and what impact this could have on the individual when asked to concentrate on different parts of the body. This quote highlights how much of a sensitive issue this could be, and particular attention is required when adapting these techniques for stroke survivors. 

“I think you have to be really mindful of the sensitivity of some of the interventions, so the body parts, if someone has paralysis of a limb and they are been told to focus on it, it could be quite emotional for them the first time they are being asked to think about this, so you have to understand that not everyone would want to do that sort of exercise…”(HCP2)

They refer to this concept again and reinforce the issue. 

“…you have also got mobility issues so if they are focusing on body parts, it might be a sensitive topic, it might be the first time they are thinking about their leg that’s got paralysis or whatever…”(HCP2)

This HCP is very mindful that by repeatedly asking a stroke survivor to focus on a specific limb it could not only lead to the survivor not completing a potentially beneficial technique but could also generate an emotional and distressing situation that should be avoided.

#### 3.1.4. Theme Four: Visual Presentation of the Intervention

The visual presentation of the techniques was raised as problematic for use with stroke survivors. This is something that needs to be carefully considered to be inclusive due to the varying nature of each individual with a stroke. The use of subtitles and key words (in bold) on the screen was included to help individuals with aphasia decipher key information and reduce the reliance on the demonstrators spoken words [38]. This was initially praised. For example, caregiver SC2 fed back that she liked the fact that it “highlighted the important words on the screen with subtitles”. Similarly, HCP1 also commented that she thought it was good how the key words were displayed when that came up. However, there were some issues regarding the subtitles matching the demonstration in that “there was one time when it didn’t match with it” (HCP1).

This was also supported by another health care professional who highlighted the importance of being very specific with what we are putting on the screen and making sure that these visual cues are appearing at the same time as the information being spoken. This is vital if individuals are using the subtitles for context and to help decipher meaning from the spoken words. 

“…you have to think that some stroke victims often have hemianopia, visual impairments, so if you are going to have bold bits on the screen, it doesn’t always correlate with what he’s saying… so for me as a listener, if I wasn’t taking in every word that he was saying, I would just think he was talking about that, but he wasn’t.”(HCP2)

During the PPI, one of the focuses was on if the amount of time to practise the interventions would be sufficient. The current intervention has a practice time of 1.5 min where the screen is blank, enabling the participant to practise that technique. What was not expected was that the visual element during that time would be problematic. For example, caregiver (SC2) discussed that when the first technique was showing in the video, when they need to practise “it is in complete darkness, there is no sound” and they “thought it stopped working”. HCP1 also expressed the same views in that it was difficult when there is that blank black screen when users “are supposed to do the exercises on their own”.

Researchers had inserted a blank screen to indicate free practice time for the techniques. It can be seen that the blank screen itself had undesired consequences by either indicating a break or looking like the video had stopped working completely. As discussed previously the choice of visual cue or wording could be problematic, but here, a complete absence of visual stimuli needs addressing.

#### 3.1.5. Theme Five: Audio Features of the Intervention

Following on from the visual elements of the intervention, there were several elements relating to the audio that required discussion. For example, SC3 did not think the sound quality of the whole video worked effectively. This stroke caregiver reveals that overall, the sound of the video appeared to not be working as well as it could have with microphone interference and crackling. They go on to raise issues with the audio features of the practice time. 

“I would want it to be consistent because a lot of times in the video its very, how to word it, jagged noises, so you would be in a relaxing moment and then there would be a bong or something like that, a sharp noise in it.”(SC3)

When the viewer is taking part in the intervention, they are presented with some time where they can practise. They are often asked to close their eyes and when it is time for them to return their attention to the demonstrator, they are presented with a loud chime or gong.

“…all of a sudden, dong, it brings you straight back out of it and you haven’t had time to do the activity”.(HCP2)

A caregiver suggests that you are brought out of that relaxed state in a way that is counterproductive to the exercise. 

“I remember this bit, just going you had to let your foot get heavy but then the microphone kept cutting out, so he would say now let your foot get heavy and it would cut out for a few seconds, then it would come crackling back in and say now let your leg get heavy, so it wasn’t a continuous erm, noise. I don’t know that bit just really irritated me”(SS4)

This was not the only audio-related element that required attention. When linking back to what SC3 states about the sound as a whole it can be seen that there are issues with the overall sound editing and sound quality of the intervention. When layering the sound over the visual elements there is a crackle every time the microphone turned back on. This is an important element to address as the intervention needs to be aphasia friendly with every effort made to ensure that it is accessible. By having unnecessary stimuli in the intervention, it is making it more difficult to learn and practise these techniques effectively. 

### 3.2. Views on the Prompts and Cues

Contributors did not discuss the use of prompts and cues in a vast amount of detail. However, what was clear is that contributors felt adding a separate video section to the intervention, where the use of prompts and cues was explained, would be beneficial.

Contributors were provided with a list of possible prompts and cues and asked if there was anything the authors had excluded. No additional prompt and cues were identified. Contributors were asked to rank their top five prompts and cues that they thought would be most beneficial to them. These results are presented in Table 2.

The results of asking contributors to rank which prompts and cues they felt would be most beneficial indicated that they preferred setting alarms and receiving email alerts from the researcher the most. Email prompts are designed to send an automated reminder from the researcher each day to prompt the individual to practise their relaxation and mindfulness. As time progresses, the individual should start to develop a routine and a behaviour change towards practising should occur, meaning the reminders are no longer necessary. This was closely followed by text message reminders and the use of post-it notes.

When discussing prompts TA talked about the possibility of using an image or photo to remind people to practise. SS2 stated that would be a good idea and “I can read, but I know so many people who had a stroke that cannot read after stroke”. This shows the need for prompts and cues to be tailored to the individuals and multiple options need to be available.

SS4 states that she uses a mindfulness app and “it pings off at 10 o’clock in the evening. Just like a little message… and it says time to wind down for bed kind of thing and then kind of hints me”. This type of prompt means that a preferred time of day can be set, and an automatic reminder is sent to the individual wherever they may be.

## 4. Discussion

The importance of conducting PPI and involving stroke survivors, stroke caregivers, and health care professionals in the design of targeted interventions has been highlighted in this paper. It has also demonstrated the feasibility of including contributors with aphasia by ensuring that PPI materials are inclusive for all stroke populations. This is vital as in previous research survivors with post-stroke aphasia are often excluded from rehabilitation studies [54]. The current PPI findings also support [50] who demonstrated that if study materials are tailored and made as accessible as possible, aphasic survivors are able to contribute to the research project. 

Furthermore, it has highlighted that as a researcher, it is easy to focus on issues that appear to need attention from an outsider’s perspective. Whereas, upon completing the PPI, the original areas of discussion were of little concern to the end users of the intervention. The length of practice time was an area the authors were keen to discuss; however, when discussing this with contributors the focus quickly shifted to new revisions that needed to be addressed as they were affecting the inclusivity and acceptability of the intervention. This directly supports the concept that effective outcomes can be achieved when the relevant expertise of the PPI contributors and the research are combined [29]. Moreover, it is crucial that the research addresses these concerns as when end users are ignored, it can result in wasted research [30]. Therefore, the focus of alteration for the current video should be centred around the concerns of the end users, whilst incorporating areas that the research team also identified as problematic. 

From the findings of this PPI, several areas of alteration have been identified. Most notably, a middle-aged stroke survivor will be the demonstrator in the modified version of the video to improve representation as well as providing a sense of ownership of the intervention to the end users. Interventions designed for stroke survivors or caregivers should be conducted by an individual from this population. This not only adds to the representation but adds ownership of the intervention to the target population. The demonstrator of a particular product, or in this case the intervention, is affiliated with the target population. For the modified intervention, that will be someone belonging to the stroke community, demonstrating techniques that have been specifically designed for them. This will act as a visual cue that the techniques can be completed by other stroke survivors, demonstrating that the current intervention has been specifically tailored to stroke survivors, which should increase uptake and engagement [31] as well possibly reducing the risk of secondary stroke [9]. 

The benefit of using a lay person and someone who is a representative of the target population may not only be down to preference. For some individuals, a clinical setting or seeing a medical professional can generate a disproportionate stress response known as the white coat effect [55]. This effect has been found to cause significant increases in blood pressure during therapist delivered aphasic speech therapy compared to self-directed therapy [56]. Moreover, Torres-Prioris et al. (2019) demonstrated that when language evaluations were administered by an individual’s husband there was a significant improvement in the severity of aphasia compared with when this was administered by a therapist [57]. These points highlight the difficulty of making decisions regarding targeted interventions, whilst also highlighting the invaluable worth of conducting PPI and working with the desired population. 

Addressing the issue of demonstrator age is difficult due to the paradox discussed in Section 3. Finding a universal demonstrator would be an impossible task. However, for the modified intervention, a middle-aged stroke survivor has been selected. Although, it is difficult to appease everyone, it is hoped that by bridging the gap between ages, a larger proportion of the population may be engaged by the intervention. It may be necessary in future interventions to provide a choice of demonstrator. 

It is important that when developing interventions and materials for stroke survivors that these visual cues are employed to ensure that they are accessible for as many individuals as possible. However, this needs to be targeted with consultation from specialists in the area to determine how this can be employed accurately and effectively. If it is not, then it could have the opposite intention, be distracting and move focus away from where it needs to be directed. This is very easy to do when able bodied designers are focusing on auditory stimuli and miss how crucial visual cues can be. Therefore, conducting PPI and actively involving end users in the development of the material is a vital practice that should be further encouraged. 

The choice of language in an intervention needs to be accessible and tailored to the target population. Moreover, a difficulty with some of the relaxation techniques is that they traditionally use abstract language, which is appropriate in the general population, but poses a linguistic problem when adapting to individuals with aphasia. Kiran et al. (2009) state that an exaggerated concreteness effect is displayed in individuals with aphasia, referring to a behavioural preference for words that are concrete compared to those that are abstract [58]. Just as visuals are important for inclusivity, the designing of interventions needs to be targeted at being aphasia friendly from the beginning of the process right down to the words chosen and the use of concrete language in the form of simple sentences. 

Moreover, the language of the specific technique needs to be carefully considered. This is seen in the body relaxation technique discussed previously. The potential for this to be distressing or uncomfortable is consistent with a finding by Merriman et al. (2015) who found that when delivering a body scan technique, individuals had variable sensations between limbs and an adaptation could include working through limbs separately [59]. In the current intervention, there is a combination of separately and together. It may be necessary for future adaptations to be implemented where specific limbs are not targeted directly and that individuals are consistently given the choice to avoid focusing on an area that makes them feel comfortable. Therefore, when designing future interventions that are stroke specific, they should consider both the physical capabilities as well as any emotional response to those physical stimuli, and the language used.

From the discussion around prompts and cues, there is a demand for a short video that explains how to include these to increase practice to the required frequency (i.e., daily). This will also introduce participants to a variety of options and allow them to pick one suited to their needs. From the discussions it is evident that there needs to be a mix of technological and non-technological options. For example, individuals could use alarm settings on their mobile phones to schedule a regular practice time. They could also place post it notes in a location that they will see frequently throughout the day that reminds them to practise their relaxation and mindfulness. These prompts can be adjusted to suit the individual and what they are doing each day. This flexibility is important as daily activities change. Practising at a regular fixed time was the least preferred out of all the prompts and cues, possibly highlighting that rigidity in set practice time was not desirable. By using these prompts and cues it will help participants practise regularly and eventually incorporate relaxation and mindfulness techniques into their daily routines. This is a method of creating sustained behaviour change through the individual’s ability to self-enact and employ behavioural change techniques individually [44,45]. 

One strength of the work (i.e., one of the process impacts) was that individual interviews were conducted. PPI contributors with communication difficulties were able to express their ideas in their own time without the pressure of other people. Moreover, one contributor used a hybrid format of speaking and then using the type function in the Teams link when they needed help expressing themselves. The information could easily be lost in a group setting if the individual is not as confident speaking in front of others. This resulted in richer data once the contributor felt comfortable sharing their views with the researcher. However, one issue was that contributors did not want to cause offence to the researcher by criticising the intervention too much. This required reassurance but it is possible that contributors withheld some points due to being polite.

Future research should aim to be as inclusive as possible with accommodations made to ensure individuals with communication difficulties can fully participate and not be excluded. In the current PPI all participant facing materials were provided in alternative formats and checked by a speech and language therapist. Moreover, materials were provided in advance to ensure contributors had sufficient time to go through them at their own pace. It was noticeable that contributors had made notes to help them express their ideas during interviews. The above adaptations are good practice and enabled members with communication difficulties to be involved and able to effectively express themselves. 

## 5. Conclusions

From the results of the PPI, several alterations have been identified which will enable the revised version to be further tailored to the needs of stroke survivors and their caregivers. By having an intervention that is specifically tailored to reduce post-stroke stress and anxiety, it is more likely that stroke survivors and caregivers will engage with the materials and achieve the desired frequency of practice, which will lead to enhanced intervention effectiveness. This could also reduce the risk of a secondary stroke [9]. 

## Figures and Tables

**Table 1 healthcare-11-00399-t001:** PPI collaborators’ demographics.

Contributor Group	I.D	Gender	Age	Ethnicity	Highest Education	Marital Status
Stroke survivor	SS1	Female	53	White British	GCSE	Divorced
Stroke survivor	SS2	Female	63	White British	A level	Married
Stroke survivor	SS3	Male	52	Chinese	Higher education	Married
Stroke survivor	SS4	Female	26	White British	Higher education	Single
Stroke Caregiver	SC1	Male	63	White British	A level	Married
Stroke Caregiver	SC2	Female	52	Chinese	Higher education	Married
Stroke Caregiver	SC3	Male	26	White British	Higher education	Single
Speech and language therapist (HCP)	HCP1	Female	53	White British	Higher education	Co-habiting
Speech and language therapist (HCP)	HCP2	Female	23	White British	Higher education	Single
Neuro physiotherapist (HCP)	HCP3	Female	41	Indian	Higher education	Married
Neuro physiotherapist (HCP)	HCP4	Female	29	Indian	Higher education	Married

**Table 2 healthcare-11-00399-t002:** Description of the four relaxation and mindfulness techniques.

Technique	Description
Body relaxation	Participants systematically focus on different parts of the body. They do not have to move but focus on the feeling of each body part.
Breath watch	Participants are guided through focusing on their breathing. Instructed not to alter their breathing but to notice their breathing coming in and out.
Thinking of a nice place	Participants are asked to think of a place where they are happy. They are asked to focus on the different senses associated with that place.
Positive emotions	Participants are asked to imagine a ball of light which fills them with happiness and warmth, generating a positive emotional experience.

**Table 3 healthcare-11-00399-t003:** Ranked preference of prompts and cues.

Prompt/Cue	Rank
Alarms	=1
Email	=1
Text	2
Post it notes	3
Stickers	4
Friends/Family	5
Link to an activity	=6
Diary	=6
Wristband	7
Smart speaker	=8
Fridge magnet	=8
Leaving a laptop open	9
Regular fixed time	10

## Data Availability

The data presented in this study are available on request from the corresponding author. The data are not publicly available due to the participants’ privacy.

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
