# Peer review of "“I Assumed It Would Be Somebody Who Had a Stroke That Was Doing This”: Views of Stroke Survivors, Caregivers, and Health Professionals on Tailoring a Relaxation and Mindfulness Intervention"

_healthcare, 2023, doi:10.3390/healthcare11030399_

Round 1

Reviewer 1 Report

Thank you for the invitation to review this manuscript. This manuscript reports patient and public involvement (PPI) in a project tailoring relaxation and mindfulness interventions for people affected by stroke. The area of study is important and addresses a priority area of research for people affected by stroke. The focus on PPI is also interesting and highlights the importance of PPI in stroke research.  

The main concern with this paper is whether the project is best described as PPI, or whether it would be better described as development work? PPI and development work share similarities, but I do wonder whether this might be better described as development work and not PPI. The reason for thinking this may be development work is because the stroker survivors, caregiver, and Health Care professionals (HCPs) provided insight and their perspectives on how the intervention might be developed/enhanced, rather than being partners in the research process. PPI purists may also have concerns about HCPs being used in this PPI activity when they do not have the same lived experience as people affected by stroke (e.g., stroke survivors and caregivers). The concern about using HCPs in PPI is also reflected in your definition of ‘public’ when describing PPI (line 34-35). The language around PPI is also important and some people may avoid using terms like ‘recruitment’ when describing partnership working with PPI members. The line between PPI and development can be blurred, but it is worth thinking about the best way of describing this work. It might be worth considering some key features of this work: anonymity of PPI members (not normally needed in PPI), ethical approval for PPI activity (not normally needed in PPI), and PPI members participation in post-intervention interviews (not common in PPI). All these elements suggest this work does not fully align with PPI work and it might be worth looking at this link for an example of similar work https://pilotfeasibilitystudies.biomedcentral.com/articles/10.1186/s40814-018-0244-1 I am not sure if this is PPI, but PPI is best reported using the GRIPP2 checklist: https://www.bmj.com/content/358/bmj.j3453 

Abstract should be structured using sub-headings (e.g., purpose, materials and methods, results, conclusions). The abstract should also add dates and locations to provide more specific content for those screening the abstract. Abstract might be better chronologically (line 19-20) (e.g., participants watched the intervention and provided feedback). 

Introduction might be better to start with clinical context (e.g., post-stroke anxiety) rather than PPI (see previous comments about PPI). It might be useful to describe and define mindfulness and relaxation for people not familiar with these approaches. This information about the interventions should differentiate between mindfulness and relaxation and provide a rationale for combining these two approaches. 

It might be useful to include effect sizes (if available) when reporting outcome data on the effectiveness of mindfulness/relaxation. It states the Lawrence review found no evidence of harm (line 78-79), but this could be due to under reporting of adverse events in the included papers (very common). There are known incidence of adverse events with mindfulness interventions and this is worth mentioning in the paper to avoid giving the false impression that mindfulness in without risks https://www.sciencedirect.com/science/article/pii/S0272735818301272  

Mindfulness and relaxation are distinct interventions. It might be worth explaining why those specific interventions were used. It might be useful to be more precise about what you mean by mindfulness and relaxation because you refer to literature on progressive muscle relaxation (line 90-91) when this is not one of the relaxation techniques used in the study. This might just be my misunderstanding but the ‘breath watch’ interventions description sounds like mindful breathing’ and body relaxation sounds like ‘body scan’. This is confusing because mindfulness breathing, and body scan are standard mindfulness techniques but are classified as a relaxation technique in this manuscript. It might be worth explaining what classification was used to select specific techniques and differentiate between mindfulness and relaxation. Just a thought, but I wonder whether it might be easier to just classify them all as mindfulness techniques, focus solely on mindfulness, and remove the confusion around relaxation techniques. 

Given part of this work is around queues and increasing engagement with and use of mindfulness/relaxation, it might be worth adding some additional content around the importance of daily practice and the link with better outcomes https://www.sciencedirect.com/science/article/pii/S0005796717300979  

Content around digital interventions (lines 107-127) might be better in the introduction/background to help provide a rationale for this work. 

The aim of the study is a little hidden and it would be better to make the aim more succinct and explicit. 

Analysis (2.6) might be worth describing the steps taken to ensure quality in Thematic Analysis. Also, might be worth citing more recent work by Braun and Clark and include reference to their evolution of their approach to Reflexive Thematic Analysis https://www.tandfonline.com/doi/abs/10.1080/2159676X.2019.1628806  

The discussion could be improved by providing more focused recommendations for practice and future research. It would also be good to consider the strengths and weaknesses of this work. 

Overall, this is an interesting and important project, and the manuscript has potential. However, there does need to be consideration given to whether this is PPI or development work and several minor revisions need making before being ready for publication.

Reviewer 2 Report

The study shed light on the view of stroke survivors, caregivers, and health care professionals on relaxation and mindfulness interventions with two aims (1) identifying the improvements in existing ways of relaxation and mindfulness techniques; and 2) acquiring patients and public involvement (PPI) consultations and preferences to ease of such daily practices. The PPI contributors were the stroke survivors and stroke caregivers, and health care professionals with a total number of 11 including 4 stroke survivors, 3 stroke caregivers, and 4 health care professionals. The intervention was about 15 minutes consisting of two mindfulness and two relaxation techniques (body relaxation, breath watch, thoughts of the peaceful place, and positive emotions). The study used thematic analysis and indicated the importance of conducting PPI for stroke survivors and stroke caregivers, and health care professionals. Moreover, the study showed the feasibility of including contributors with aphasia by ensuring that PPI materials are inclusive for all stroke populations. The study concluded that the results of the PPI indicated that the revised version could be tailored to the need of stroke survivors and caregivers with the aim of reducing post-stroke stress and anxiety. Here are some minor comments:

Comment 1: please discuss the impact of sex and gender on the outcome of the study.

Comment 2: Please modify the abstract and write the aim and the conclusion of the study in the more clear writing

Comment 3: The manuscript is long. This reviewer believes that both the introduction and discussion could be rewritten shorter and more clearly/concisely.
